# Comparison of outcomes between video laryngoscopy and flexible fiberoptic bronchoscopy for endotracheal intubation in adults with cervical neck immobilization: A systematic review and meta-analysis of randomized controlled trials

Nana Guo[1], Xuxin Wen[1], Xiao Wang[1], Junling Yang[2], Haidong Zhou[2], Jianli Guo[2], Yun Su[1]*, Tingxin Zhang[3]*

1 Critical Care Medicine, Ordos Central Hospital, Ordos, China, 2 Anesthesiology Department, Ordos Central Hospital, Ordos, China, 3 Department of Orthopedics, Ordos Central Hospital, Ordos, China

* suyunicu@163.com (YS); 2968176466@qq.com (TZ)

## Abstract

### Purpose

Comparing the outcomes of video-laryngoscopy and flexible fiberoptic bronchoscopy for endotracheal intubation in patients with cervical spine immobilization

### Methods

All of the comparative studies published in the PubMed, Cochrane Library, Medline, Web of Science, and EMBASE databases as of 8 Jan 2024 were included. All outcomes were analyzed using Review Manager 5.4. The primary outcomes were the successful first-attempt intubation rate, intubation time, heart rate after intubation, mean arterial pressure after intubation, overall intubation success rate, risk of tissue damage and sore throat.

### Results

The meta-analysis included six randomized controlled studies with a total of 694 patients. The outcomes of the meta-analysis revealed that the use of video laryngoscopy was better than flexible fiberoptic bronchoscopy in terms of the successful first-attempt intubation rate (P<0.05) and intubation time (P<0.05) in patients with cervical spine immobilization. However, there were no statistically significant differences in heart rate after intubation, mean arterial pressure after intubation, overall intubation success rate, risk of tissue damage, or sore throat (all P>0.05) between the video laryngoscopy and flexible fiberoptic bronchoscopy groups.

**Data Availability Statement:** All relevant data are within the manuscript and its Supporting Information files.

**Funding:** The author(s) received no specific funding for this work.

**Competing interests:** The authors have declared that no competing interests exist.

## Conclusions

Compared with flexible fiberoptic bronchoscopy, video laryngoscopy has superior tracheal intubation performance in terms of the first-attempt success rate and intubation speed. This finding was observed in patients with cervical spine immobilization who utilized a cervical collar to simulate a difficult airway. Additionally, both types of scopes demonstrated similar complication rates. Current evidence suggests that video laryngoscopy is better suited than flexible fiberoptic bronchoscopy for endotracheal intubation in patients immobilized with a cervical collar.

## Trial registration

Systematic review protocol: CRD42024499868.

## 1. Introduction

Endotracheal intubation in patients with cervical spine instability requires special attention to prevent excessive motion of the cervical spine, which could result in secondary neurological damage [1,2]. Managing the airway in patients with suspected cervical instability poses a challenge for anesthesiologists. In trauma patients, the airway is typically secured through direct laryngoscopy. However, this procedure can cause movement of the cervical spine, potentially worsening existing injuries. To minimize the risk of spinal cord nerve damage during tracheal intubation, immobilization of the cervical spine using a cervical collar is recommended [3,4]. However, cervical collar immobilization may increase the difficulty of direct laryngoscopic tracheal intubation because it limits the alignment of the three airway axes. Therefore, video laryngoscopy (VL) and flexible fiberoptic bronchoscopy (FFB) may be used to increase the success rate of tracheal intubation in patients with cervical spine immobilization [5]. Because they reduce cervical spine mobility in one step compared with direct laryngoscopy, they provide an additional safety measure for preventing the aggravation of preexisting injuries [6]. However, selecting the appropriate intubation device remains a clinical challenge because of the unique advantages and disadvantages associated with each device. For example, using FFB often necessitates techniques to expand the retropharyngeal space, potentially resulting in varying degrees of cervical spine movement [7]. Moreover, the use of FFB requires additional specialized training [8]. While VL results in less cervical extension than direct laryngoscopy does, it may take longer to perform the intubation procedure [9].

In this context, several studies [10–15] have compared video laryngoscopy and fiberoptic endotracheal intubation in patients with cervical spine immobilization. However, all previously published studies have obvious limitations, such as insufficient sample sizes and failure to clarify the specific differences between VLs and FFBs. There is currently insufficient level 1 evidence to demonstrate specific differences between VL and FFB during intubation in patients immobilized with a cervical collar. Therefore, we reviewed previous randomized controlled trials and performed a meta-analysis to synthesize the evidence.

## 2. Methods

### 2.1. Literature search strategy

Systematic literature searches were conducted in five electronic databases (PubMed, Cochrane Library, Medline, Web of Science, and EMBASE) using a combination of MeSH (Medical

Subject Heading) terms and free text words: "video laryngoscopy", "flexible fiberoptic bronchoscopy" and "cervical collar". The search was performed from the time the databases were built until 8 Jan 2024, without any language or publication year restrictions. Additionally, the bibliographies of RCTs, meta-analyses, and systematic reviews were manually searched to ensure comprehensive coverage. See S1 Checklist.

## 2.2. Selection of studies

The study inclusion and exclusion processes were conducted in two groups. Initially, the selection was based on the title and abstract. If a decision could not be reached based on the summary, the full text of the article was retrieved. In cases where there was a disagreement between the two groups, the selection committee discussed the article until a consensus was reached. See S1 Table for a numbered table of all studies.

## 2.3. Inclusion and exclusion criteria

We included studies that met the following criteria: (1). RCTs; (2). comparative study on the efficacy of VL and FFB in endotracheal intubation; and (3). comparison outcomes included at least one of the following: successful first-attempt intubation rate, intubation time, heart rate after intubation, mean arterial pressure after intubation, overall intubation success rate, risk of tissue damage, and sore throat. Studies were excluded if they met any of the following criteria: (1). editorials, letters, reviews, case reports, and cadaver or animal experiments; (2). a history of difficult tracheal intubation; (3). did not meet the inclusion criteria; and (4). data of the comparison outcomes could not be extracted.

## 2.4. Data extraction

Two reviewers used standardized data extraction tables. The extracted data included authors, publication date, title, country, study design, number of patients, mean age of patients, neck fixation technique, and comparison outcomes. The comparison outcomes included successful first-attempt intubation rate, intubation time, heart rate after intubation, mean arterial pressure after intubation, overall intubation success rate, risk of tissue damage, and sore throat. All the data were extracted from article texts, tables, and figures. The research author was contacted for missing data or further information. Two reviewers independently extracted the data; differences were resolved through discussion, and a consensus was reached by discussion with third parties. The data extraction outcomes are shown in Table 1. See S2 Table for all data extracted.

## 2.5. Data analysis

We used Review Manager Version 5.4 (Copenhagen: The Nordic Cochrane Centre, The Cochrane Collaboration) to analyze the data of all outcomes and compare the VL group with the FFB group. For continuous outcomes, such as the intubation time, heart rate after intubation and mean arterial pressure after intubation, the means and standard deviations were pooled into a weighted mean difference (WMD) and 95% confidence interval (CI). Risk ratios (RRs) and 95% CIs were used to evaluate dichotomous outcomes, such as the successful first-attempt intubation rate, overall intubation success rate, risk of tissue damage and sore throat. We used $I^2$ to quantify heterogeneity. If $I^2 > 50\%$, the heterogeneity was significant, and the unstandardized mean difference was estimated using a random effects model. Otherwise, a fixed-effects model was applied.

Table 1. Characteristics of included studies.

| Author (years) | country | Study type | Number of Samples VL/FFB | Gender (male) VL /FFB | Average age VL /FFB | BMI (kg/m²) | Neck fixation technique | Outcomes |
|---|---|---|---|---|---|---|---|---|
| Abdullah (2013) [10] | Singapore | RCT | 30/30 | - | 43.8/49.9 | 23.4/25.7 | cervical collar | 1,2,5–7 |
| Yumul (2014) [11] | USA | RCT | 70/70 | 38/38 | 55/50 | 27/28 | cervical collar | 1–7 |
| Wahba (2012) [12] | Egypt | RCT | 25/25 | 21/20 | 37/34 | 26.9/29.0 | cervical collar | 1–5,7 |
| Shulman (2001) [13] | USA | RCT | 25/25 | - | 44/44 | - | cervical collar | 1,2,5 |
| Choi (2023) [14] | Korea | RCT | 166/164 | 101/114 | 56.4/56.6 | 24.5/24.6 | cervical collar | 1–3,5–7 |
| Gill (2015) [15] | India | RCT | 32/32 | 19/19 | 36.4/37.4 | - | cervical collar | 1,2,4,5 |

*Outcomes*:1. Successful first-attempt intubation rate,2. Intubation time,3. Heart rate after intubation,4. Mean arterial pressure after intubation,5. Overall intubation success rate,6. Risk of tissue damage,7. Sore throat.

VL: Video laryngoscopy FFB: Flexible fiberoptic bronchoscopy RCT:Randomized controlled trial.

### 2.6. Quality assessment

For RCTs, we followed the guidelines outlined in the Cochrane Handbook for Systematic Reviews of Interventions [16], specifically focusing on 7 domains: random sequence generation, allocation concealment, blinding of participants and personnel, blinding of outcome assessment, incomplete outcome data, selective outcome reporting, and other sources of bias (Fig 1). The quality assessment was independently conducted by two reviewers, and any disagreements were resolved through discussion with a third party. See S3 Table for Cochrance Quality Assessment Scale.

## 3. Results

### 3.1. Literature search

The study selection and exclusion process for the current meta-analysis is shown in (Fig 2). We searched 642 studies from 5 electronic databases. Of these, 606 studies were excluded because of duplication (n = 55) or irrelevance (n = 551). After careful full-text evaluation, 6 studies [10–15] were reviewed, and the data were extracted. The demographic and clinical characteristics of the 6 studies are described in Table 1. A total of 348 patients underwent VL intubation, while 346 patients underwent FFB intubation. All patients used a cervical collar to fix the cervical spine. Six studies [10–15] reported intubation time, successful first-attempt intubation rate, and overall intubation success rate. Three studies [11,12,15] reported mean arterial pressure after intubation. Heart rate after intubation was reported in 3 studies [11,12,14]. Postintubation sore throat was reported in 4 studies [10–12,14]. The risk of tissue damage was reported in 3 studies [10,11,14].

### 3.2. Intubation time

Six studies [10–16] with a total of 694 patients (VL group, n = 348 vs. FFB group, n = 346) compared the mean intubation time. The meta-analysis indicated that the VL group had significantly fewer intubation times than the FFB group (WMD, -27.07; 95% CI, -34.01 to -20.13; $P<0.05$). The heterogeneity test outcome ($I^2$ = 94%) indicated significant heterogeneity (Fig 3).

### 3.3. Successful first-attempt intubation rate

Six studies [10–16] with a total of 694 patients (VL group, n = 348 vs. FFB group, n = 346) compared the successful first-attempt intubation rate. The pooled outcomes indicated that the

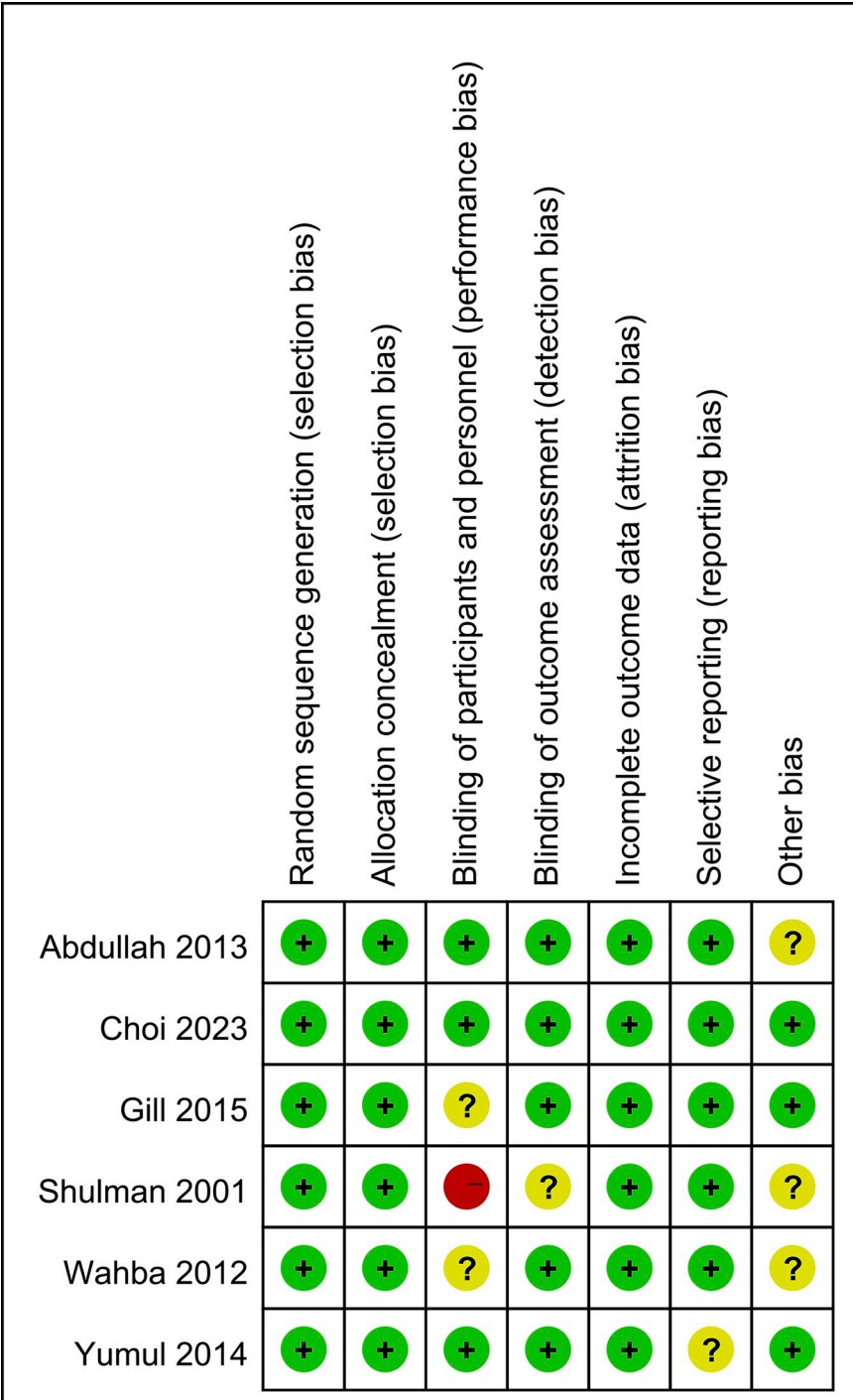

**Fig 1. Risk of bias summary.** All studies considered to have a low overall risk of bias.

FFB group had a significantly lower first-attempt intubation rate than the VL group did (RR, 3.28; 95% CI, 1.27 to 8.47; P<0.05). The heterogeneity test outcome ($I^2$ = 52%) indicated significant heterogeneity (Fig 4).

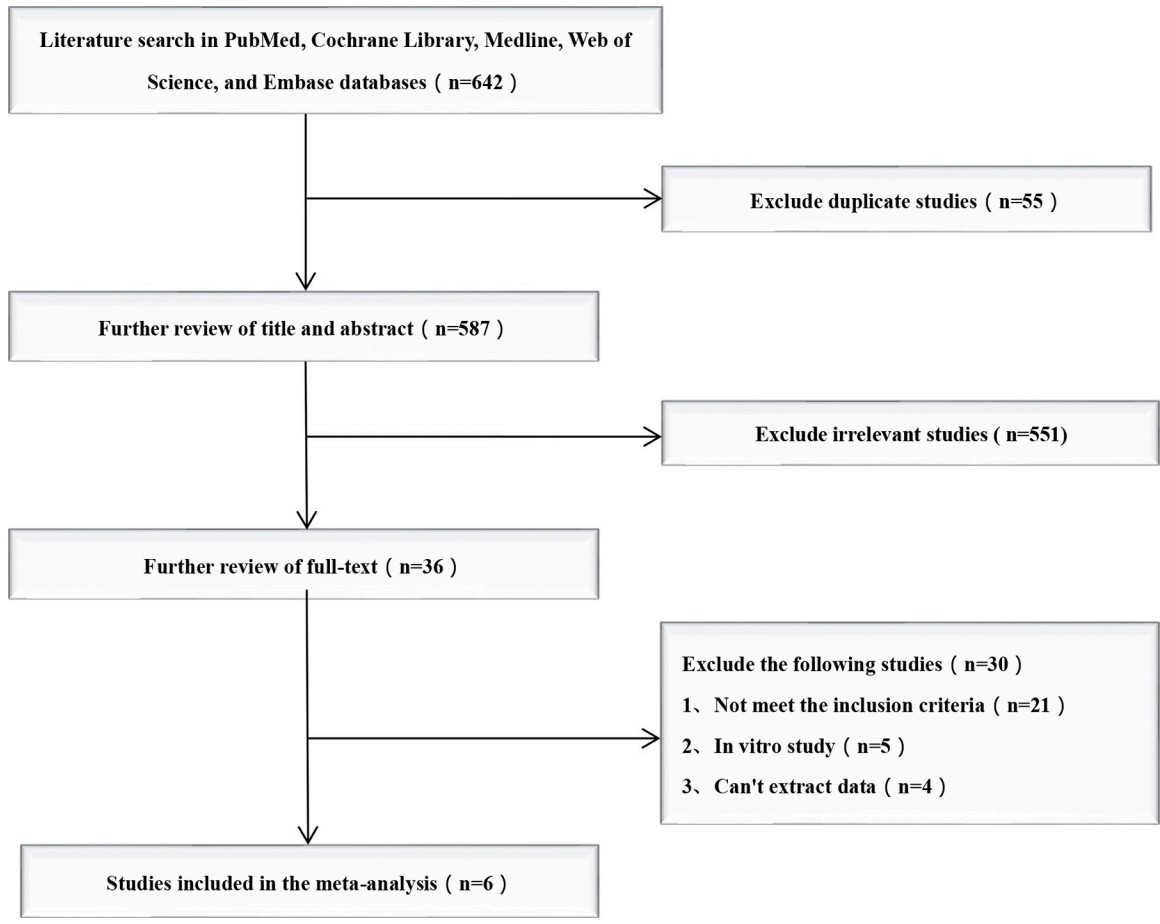

**Fig 2. Flow diagram of study selection.**

### 3.4. Overall intubation success rate

Six studies [10–16] with a total of 694 patients (VL group, n = 348 vs. FFB group, n = 346) compared the overall intubation success rate. The meta-analysis indicated no significant differences between the VL and FFB groups(RR, 1.01; 95% CI, 0.99 to 1.02; P>0.05). The heterogeneity test outcome was $I^2 = 0$, and the fixed effects model was applied (Fig 5).

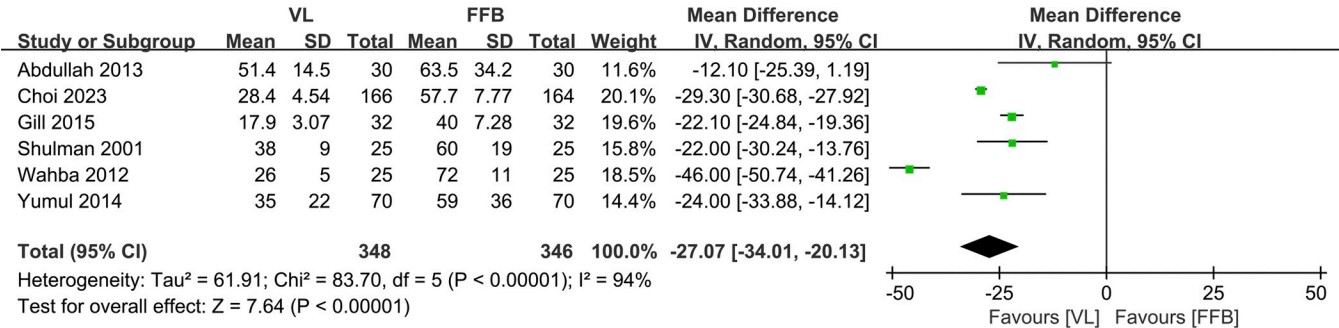

**Fig 3. Meta-analysis of VL group versus FFB group in intubation time.**

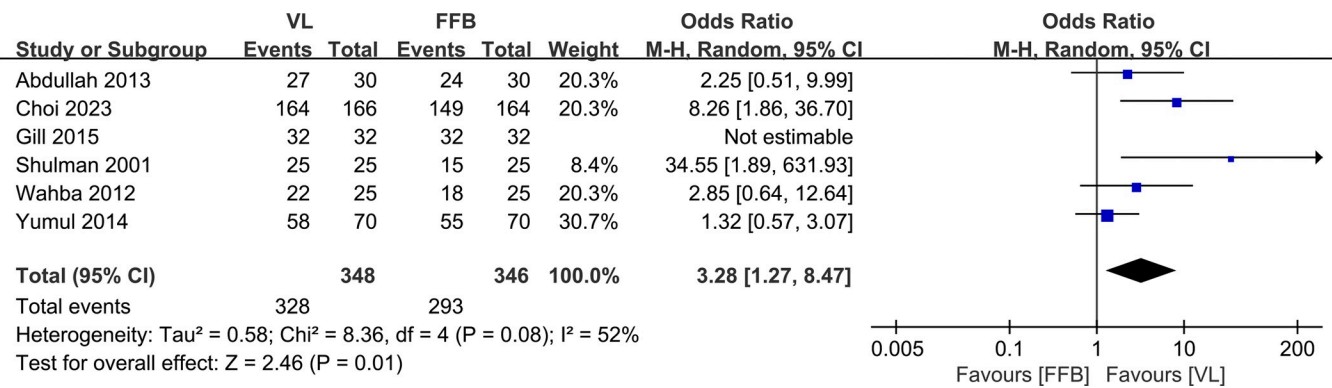

**Fig 4. Meta-analysis of VL group versus FFB group in successful first-attempt intubation rate.**

### 3.5. Mean arterial pressure after intubation

Three studies [11,12,15] with a total of 254 patients (VL group, n = 127 vs. FFB group, n = 127) compared the mean arterial pressure after intubation. The meta-analysis indicated no significant differences between the VL and FFB groups (WMD, 1.75; 95% CI, -12.76 to 16.26; P>0.05). The heterogeneity test outcome ($I^2$ = 96%) indicated significant heterogeneity (Fig 6).

### 3.6. Heart rate after intubation

Three studies [11,12,14] with a total of 520 patients (VL group, n = 261 vs. FFB group, n = 259) compared the mean arterial pressure after intubation. The meta-analysis indicated no significant differences between the VL and FFB groups (WMD, 6.32; 95% CI, -7.10 to 19.74; P>0.05). The heterogeneity test outcome ($I^2$ = 97%) indicated significant heterogeneity (Fig 7).

### 3.7. Risk of tissue damage

Three studies [10,11,14] with a total of 530 patients (VL group, n = 266 vs. FFB group, n = 264) compared the rate of tissue damage. The meta-analysis concluded no significant differences between the VL and FFB groups (RR, 0.87; 95% CI, 0.68 to 1.11; P>0.05). The heterogeneity test outcome was $I^2$ = 0, and the fixed effects model was applied (Fig 8).

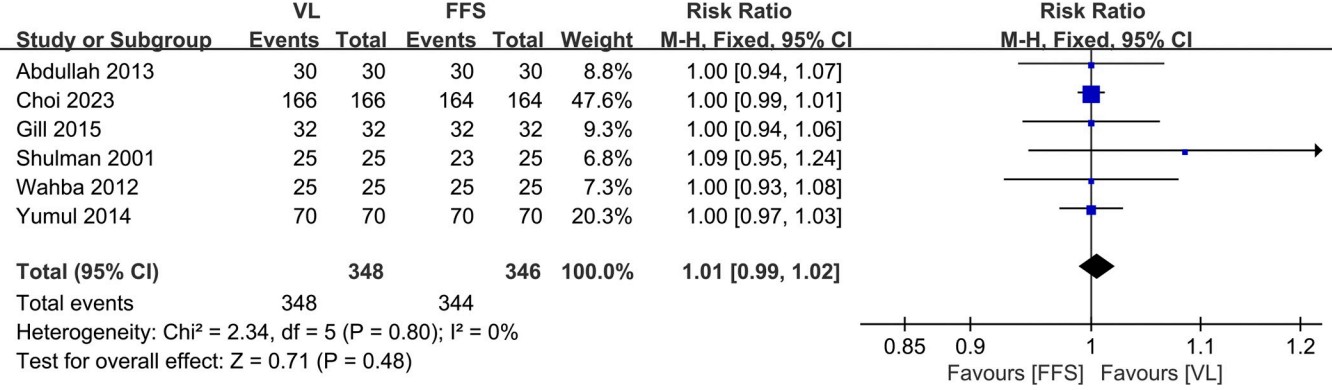

**Fig 5. Meta-analysis of VL group versus FFB group in overall intubation success rate.**

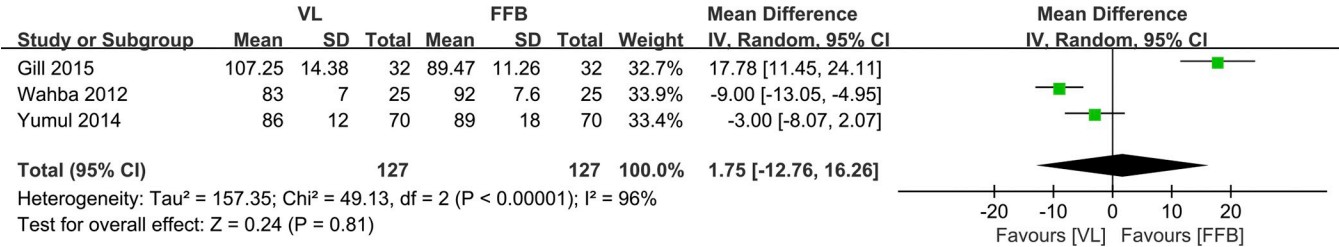

**Fig 6. Meta-analysis of VL group versus FFB group in mean arterial pressure after intubation.**

### 3.8. Sore throat

Four studies [10–12,14] with a total of 580 patients (VL group, n = 291 vs. FFB group, n = 289) compared the rate of sore throat. The meta-analysis concluded no significant differences between the VL and FFB groups (RR, 1.04; 95% CI, 0.82 to 1.32; P>0.05). The heterogeneity test outcome was $I^2 = 0$, and the fixed effects model was applied (Fig 9).

## 4. Discussion

Accidents, such as vehicle accidents and falls from heights, can result in cervical spine injuries. For patients with cervical spine instability, manual linear fixation or cervical collar fixation is often used to maintain the neutral position of the cervical spine and prevent flexion and rotation, thus avoiding secondary nerve damage [17]. While manual linear fixation and cervical collar fixation are simple and easy to implement, it is important to note that cervical spine fixation can have negative effects on intubation conditions, limit the patient's mouth opening, and significantly increase the glottis grade exposed by the laryngoscope [18]. This can transform a normal airway into a difficult airway, potentially leading to hypoxic events and endangering the patient's life. The safest way to deal with this type of patient is FFB tracheal intubation under awake topical anesthesia, but FFB has a long learning curve and is difficult to master clinically [19]. Different from FFB, VL has a shorter learning curve, which is very conducive to clinical promotion and application.

Clinically, it is very important that endotracheal intubation is successful on the first attempt. Failure of the first tracheal intubation will lead to a lower success rate of subsequent intubation, even if another anesthesiologist is changed or other equipment is used for intubation [20,21]. Furthermore, multiple endotracheal intubation attempts are known to increase the likelihood of adverse events. Therefore, it is necessary to determine which endotracheal intubation device facilitates first-time intubation success, especially in patients whose trachea may be difficult to intubate with cervical spine immobilization. Although some studies have

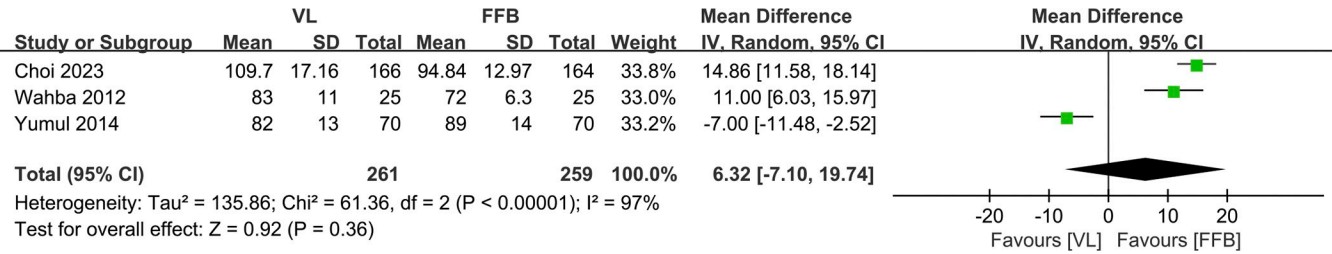

**Fig 7. Meta-analysis of VL group versus FFB group in heart rate after intubation.**

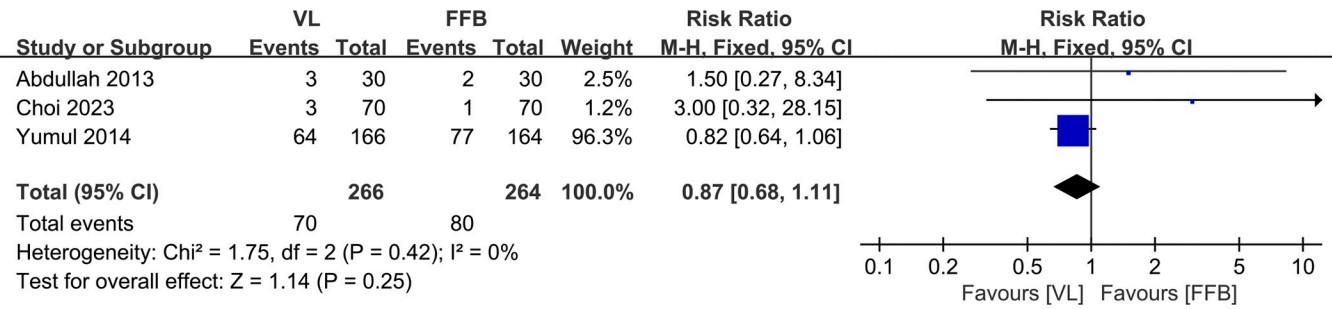

**Fig 8. Meta-analysis of VL group versus FFB group in risk of tissue damage.**

shown that the intubation time of VL is shorter than that of FFB [10–13], it is still controversial whether the first intubation success rate of VL is higher than that of FFB.

To our knowledge, there has been no meta-analysis specifically comparing the outcomes of tracheal intubation between VL and FFB in patients with cervical spine immobilization. In our meta-analysis, we extracted information from 6 published RCTs using the Cochrane Handbook for Systematic Reviews of Interventions to assess the quality of 694 patients. The outcomes of our analysis indicated that the included literature was of high quality, all studies considered to have a low overall risk of bias (Fig 1). Our study demonstrated that patients with cervical collar fixation had a higher first intubation success rate in the VL group compared to the FFB group. Additionally, the intubation time was shorter in the VL group. However, there were no significant differences between the VL and FFB groups in terms of overall intubation success rate, heart rate after intubation, mean arterial pressure after intubation, tissue damage, and incidence of sore throat.

Our analysis yielded results that were consistent with the findings in previous studies [10–15] comparing VL and FFB intubation in patients with cervical spine immobilization regarding endotracheal intubation time. For patients with cervical collar fixation, VL may result in shorter intubation times compared to FFB for two reasons. First, VL provides a view of both the larynx and the intubation process, allowing for direct visualization and reducing the need for contact with laryngeal structures, thus potentially reducing the time required for intubation. Additionally, the blade in VL can be used to open the narrow oropharyngeal space, facilitating the exposure of airway structures such as the epiglottis and further contributing to a shorter intubation time.

Regarding the successful first-attempt intubation rate in patients with cervical spine immobilization, Abdullah et al. [10] reported that there was no difference in the rate of successful

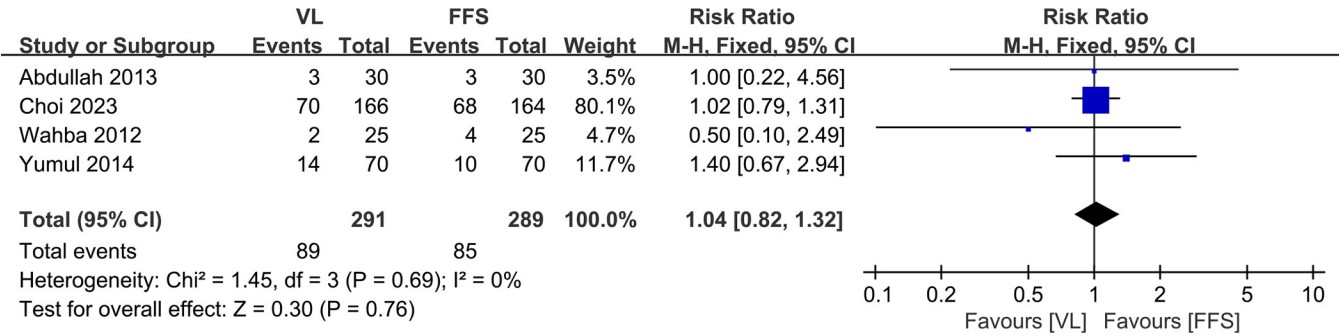

**Fig 9. Meta-analysis of VL group versus FFB group in sore throat.**

intubation between VL and FFB groups. However, Choi et al. [14] reported that compared with FFB, VL had better tracheal intubation performance in terms of the first attempt success rate. We also showed that for patients with cervical collar immobilization, the first-time intubation success rate using VL was higher than that of FFB. While FFB endotracheal intubation allows for a direct view of the larynx, the actual intubation process cannot be viewed on video. To overcome any resistance encountered during the advancement of the tracheal tube, it is necessary to rotate the tube around the axis of the FFB. This rotation increases the difficulty of the intubation procedure to some extent, resulting in a lower success rate for first-time intubation. In addition, most studies do not have clear definitions and criteria for first-time intubation failure, and future studies need to further clarify and unify the criteria.

Intubation is well-known to elicit an amplified hemodynamic response characterized by tachycardia, hypertension, and dysrhythmias. A study conducted by McCoy et al. [22] revealed that the use of McCoy laryngoscope reduces the force needed for laryngoscopy by elevating the epiglottis, thereby mitigating the stress response associated with laryngoscopy. Adachi et al. [23] compared FFB intubation with conventional direct laryngoscopy and observed stable hemodynamics in the FFB group. The study by Gill et al. [15] showed that FFB was superior to VL in terms of stable hemodynamic response to intubation and glottis exposure. However, our results showed that there were no significant differences in heart rate after intubation and mean arterial pressure after intubation between the VL group and the FFB group.

One limitation of this meta-analysis is that the studies included did not involve actual trauma patients and did not utilize rapid sequence induction procedures. As a result, the findings of this study may not apply to all patients with cervical spine immobilization in real-life situations. Furthermore, it is possible that the unfamiliarity of clinicians with FFB intubation techniques, in comparison to VL, could have introduced bias into our findings.

## 5. Conclusion

According to our analysis, the use of VL intubation in patients with cervical spine immobilization can shorten the intubation time and increase the success rate of first-time intubation, compared with FFB intubation. However, the clinical results of FFB and VL were similar in terms of overall intubation success rate, heart rate after intubation, mean arterial pressure after intubation, tissue damage, and incidence of sore throat. Based on the current evidence, we recommend prioritizing the use of video laryngoscopy in patients with cervical spine trauma to shorten the intubation time and increase the success rate of first-time intubation.

## Supporting information

**S1 Checklist. PRISMA 2020 checklist.**
(DOCX)

**S1 Table. A numbered table of all studies.**
(DOCX)

**S2 Table. All data extracted.**
(XLSX)

**S3 Table. Cochrance quality assessment scale.**
(XLSX)

## Acknowledgments

We would like to thank the authors of the included studies for their contributions to this systematic review and meta-analysis study.

## Author Contributions

**Data curation:** Nana Guo, Xiao Wang, Jianli Guo, Yun Su, Tingxin Zhang.

**Formal analysis:** Xiao Wang, Haidong Zhou, Jianli Guo, Yun Su, Tingxin Zhang.

**Methodology:** Nana Guo, Junling Yang, Yun Su, Tingxin Zhang.

**Project administration:** Junling Yang.

**Resources:** Xuxin Wen, Haidong Zhou.

**Software:** Xuxin Wen, Yun Su, Tingxin Zhang.

**Validation:** Haidong Zhou.

**Writing – original draft:** Tingxin Zhang.

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
