## [Decision Letter · Decision Letter 0]

20 Sep 2024

PONE-D-24-27464Comparison of outcomes between video-laryngoscope and flexible fiberoptic bronchoscope for endotracheal intubation in adults with cervical neck immobilisation:A systematic review and meta-analysis of randomised controlled trialsPLOS ONE

Dear Dr. Zhang,

Thank you for submitting your manuscript to PLOS ONE. After careful consideration, we feel that it has merit but does not fully meet PLOS ONE’s publication criteria as it currently stands. Therefore, we invite you to submit a revised version of the manuscript that addresses the points raised during the review process.

We look forward to receiving your revised manuscript.

Kind regards,

Ilker Kacer, Assoc. Prof. M.D.

Academic Editor

PLOS ONE

Journal Requirements:

3. As required by our policy on Data Availability, please ensure your manuscript or supplementary information includes the following: 

Additional Editor Comments:

Thank you for your valuable article.

Acceptance is avaiblable after minor grammar and writing errors.

Reviewers' comments:

Reviewer's Responses to Questions

**Comments to the Author**

1. Is the manuscript technically sound, and do the data support the conclusions?

Reviewer #1: Yes

Reviewer #2: Yes

2. Has the statistical analysis been performed appropriately and rigorously? 

Reviewer #1: Yes

Reviewer #2: Yes

3. Have the authors made all data underlying the findings in their manuscript fully available?

Reviewer #1: Yes

Reviewer #2: Yes

4. Is the manuscript presented in an intelligible fashion and written in standard English?

Reviewer #1: Yes

Reviewer #2: No

5. Review Comments to the Author

Reviewer #1: The decision to address a topic that has not been previously subjected to meta-analysis has proven to be a fruitful one. Nevertheless, when comparing endotracheal intubation methods, it is essential to consider patient-related factors, if present in the included study, as they may influence the efficacy of the procedure. In the absence of patient-related factors in the included studies, it is essential to include this information in the meta-analysis. The evaluation of the effectiveness of the methods being compared will be enhanced by the inclusion of patient-related factors.

Reviewer #2: 1- I consider this manuscript technically sound, and the data supports conclusions.

2- I believe the statistical analysis was performed appropriately and rigorously.

3- Yes

4- I believe the manuscript has some minor grammar and writing errors.

5- I consider this study worth publishing and will make an important merit to the literature

6. PLOS authors have the option to publish the peer review history of their article (what does this mean?). If published, this will include your full peer review and any attached files.

Reviewer #1: No

Reviewer #2: No

---

## [Author Response · Author response to Decision Letter 0]

18 Oct 2024

Responses to the comments of Reviewer #1

Thank you for taking the time to read and comment on our manuscript.

1. Reviewer #1: The decision to address a topic that has not been previously subjected to meta-analysis has proven to be a fruitful one. Nevertheless, when comparing endotracheal intubation methods, it is essential to consider patient-related factors, if present in the included study, as they may influence the efficacy of the procedure. In the absence of patient-related factors in the included studies, it is essential to include this information in the meta-analysis. The evaluation of the effectiveness of the methods being compared will be enhanced by the inclusion of patient-related factors.

Thank you for your comments and suggestions. When comparing endotracheal intubation methods, the presence of patient-related factors in the included studies may influence the outcomes of the procedure. In our meta-analysis, we did not incorporate these patient-related factors, which represents a limitation of this study. In future research, we aim to improve the evaluation of the effectiveness of the compared methods by including patient-related factors. This presents a new direction for subsequent investigations.

Responses to the comments of Reviewer #2

Thank you for your recognition of our paper.

2. Reviewer #2 I consider this manuscript technically sound, and the data supports conclusions. I believe the statistical analysis was performed appropriately and rigorously.I believe the manuscript has some minor grammar and writing errors. I consider this study worth publishing and will make an important merit to the literature.

Thank you for your valuable comments. For grammatical and writing errors in the manuscript, we have asked professional English editors from AJE to correct them to avoid errors in language, grammar, punctuation and spelling. The verification code for editing the certificate is: F2F6-4F68-83FD-BFA1-D650.

---

## [Editor Report · Decision Letter 1]

22 Oct 2024

Comparison of outcomes between video laryngoscopy and flexible fiberoptic bronchoscopy for endotracheal intubation in adults with cervical neck immobilization: A systematic review and meta-analysis of randomized controlled trials

PONE-D-24-27464R1

Dear Dr. Zhang,

We’re pleased to inform you that your manuscript has been judged scientifically suitable for publication and will be formally accepted for publication once it meets all outstanding technical requirements.

Kind regards,

Ilker Kacer, Assoc. Prof. M.D.

Academic Editor

PLOS ONE
---

## [Editor Report · Acceptance letter]

6 Nov 2024

PONE-D-24-27464R1 

PLOS ONE

Dear Dr. Zhang, 

I'm pleased to inform you that your manuscript has been deemed suitable for publication in PLOS ONE. Congratulations! Your manuscript is now being handed over to our production team.

Kind regards, 

on behalf of

Mr. Ilker Kacer 

Academic Editor

PLOS ONE